# Closing the Curious Case of Neural Text Degeneration

**Matthew Finlayson**[*]
University of Southern California

**John Hewitt**
Stanford University

**Alexander Koller**
Saarland University

**Swabha Swayamdipta**
University of Southern California

**Ashish Sabharwal**
The Allen Institute for AI

## ABSTRACT

Despite their ubiquity in language generation, it remains unknown why truncation sampling heuristics like nucleus sampling are so effective. We provide a theoretical explanation for the effectiveness of the truncation sampling by proving that truncation methods that discard tokens below some probability threshold (the most common type of truncation) can guarantee that all sampled tokens have nonzero true probability. However, thresholds are a coarse heuristic, and necessarily discard some tokens with nonzero true probability as well. In pursuit of a more precise sampling strategy, we show that we can leverage a known source of model errors, the softmax bottleneck, to prove that certain tokens have nonzero true probability, without relying on a threshold. Based on our findings, we develop an experimental truncation strategy and the present pilot studies demonstrating the promise of this type of algorithm. Our evaluations show that our method outperforms its threshold-based counterparts under automatic and human evaluation metrics for low-entropy (i.e., close to greedy) open-ended text generation. Our theoretical findings and pilot experiments provide both insight into why truncation sampling works, and make progress toward more expressive sampling algorithms that better surface the generative capabilities of large language models.

## 1 INTRODUCTION

Crucial to the remarkable generative capabilities of today's large language models (LLMs) (OpenAI, 2023; Touvron et al., 2023; Chowdhery et al., 2022) are the sampling algorithms responsible for selecting the next token at each timestep. The most common of these algorithms use a simple truncation strategy: sample only the tokens that have probability greater than some threshold (Holtzman et al., 2020; Fan et al., 2018). In the quest for high-entropy generation wherein one wants to be able to generate multiple good completions, it has been empirically established that the search for the highest-likelihood strings through e.g., beam search or greedy decoding led to low-quality generations (Hashimoto et al., 2019). Threshold-based truncation sampling presents a compelling alternative: by avoiding the tokens at the tail end of the distribution which correspond to degenerate text it produces significantly more coherent generations (Ippolito et al., 2019; Holtzman et al., 2020; DeLucia et al., 2021). However, beyond the intuition that language models tend to assign too much probability to tokens that should have 0 or near-0 probability (akin to smoothing (Hewitt et al., 2022)), prior work has been limited in establishing *why* truncation sampling is so essential in autoregressive generation.

In this paper, we provide a precise mathematical explanation to elucidate the extraordinary success of threshold-based truncation sampling (§3). First, we prove via an argument about log-probability errors that threshold sampling is guaranteed to only sample tokens in the support of the true distribution, so long as the chosen threshold is larger than some bound (Corollary 1). Next, we present a method to more directly account for a likely source of tail errors: the *softmax bottleneck* (Yang et al., 2018), which states that the low-rank softmax matrix used at the output layer of language models causes probability errors in the model's output distribution (§4). Specifically, we show how to leverage

---

[*]Corresponding author `mfinlays@usc.edu`

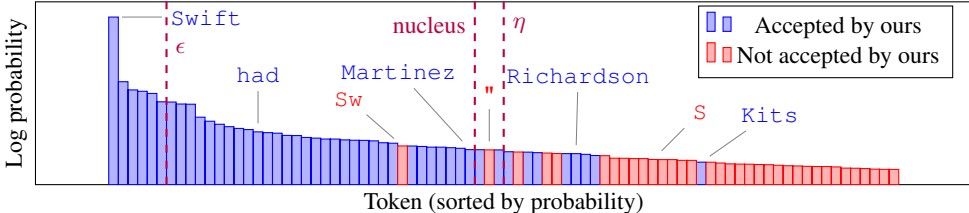

Figure 1: The next-token distribution from GPT-2 XL for the prefix "`Taylor`", with the tokens ordered by probability. Dashed vertical lines denote thresholds used to reject low-probability tokens, under various truncation strategies. Our basis-aware-threshold (BAT) sampling accepts tokens shown in blue and rejects those in orange. As evident, BAT rejects some implausible tokens assigned high probability under the model while accepting many plausible yet low-probability tokens—this is not possible under truncation sampling. BAT uses the softmax matrix to find tokens that might have non-zero true probability, without relying on a threshold. See more examples in Fig. 4.

the restricted structure imposed by the softmax bottleneck to more precisely determine (relative to threshold-based truncation) which tokens are in the support of the true distribution (Theorem 2). At a high level, the idea is to declare a token to be in the support if its probability is nonzero not only in the predicted distribution but also in *all* distributions that are "similar" to it (in a precise technical sense) from the perspective of the softmax matrix. This presents a more nuanced strategy compared to threshold-based truncation sampling: our algorithm does not rely on a threshold, thereby allowing higher probability tokens to be discarded while keeping some lower-probability tokens.

We conduct a pilot investigation (§5) to empirically evaluate this basis-aware truncation sampling approach. Our results shows improvements on an open-ended generation task via both automatic and human evaluation metrics under low-entropy generation (i.e., close to greedy). Figure 1 illustrates our algorithm's more nuanced token selection strategy qualitatively (also see Figure 4). Unlike threshold-based truncation methods (each shown with a dotted vertical line), our method can selectively discard low-quality tokens while still keeping high-quality but lower-probability tokens. This is accomplished by taking into account linear dependencies between token embeddings.[1]

Overall our work provides theoretical insights which motivate a practical method and show how truncation sampling avoids errors in a language model by mitigating the softmax bottleneck.

## 2 BACKGROUND

**Autoregressive language models** Autoregressive language models are trained as next-word-predictors: given a prefix, the model assigns a probability to each token in a vocabulary of size $v$ as a prediction of which token comes next. Given an input prefix, a model produces a vector $\boldsymbol{h} \in \mathbb{R}^d$, which we refer to as the *hidden state*, and hyperparameter $d$ as the *hidden size*. The model then uses a linear map with matrix $\boldsymbol{W} \in \mathbb{R}^{v \times d}$ to obtain logits $\boldsymbol{W}\boldsymbol{h} \in \mathbb{R}^v$, to which it applies the softmax function to obtain a probability distribution over tokens in the vocabulary:

$$\hat{\boldsymbol{p}} = \mathrm{softmax}(\boldsymbol{W}\boldsymbol{h}) = \frac{\exp(\boldsymbol{W}\boldsymbol{h})}{\sum_{i=1}^{v} \exp(Wh)_i}.$$

The matrix $\boldsymbol{W}$ is commonly referred to as the *softmax matrix* because it is applied directly before the softmax, or the *embedding matrix*. Generally models are trained to output the $\hat{\boldsymbol{p}}$ that minimizes the cross entropy with the conditional true distribution[2] $\boldsymbol{p}^*$: $\mathrm{crossentropy}(\boldsymbol{p}^*, \hat{\boldsymbol{p}}) = \sum_{i=1}^{v} p_i^* \log \hat{p}_i$.

**Generation via truncation sampling** Language models can autoregressively generate text by sampling a token from $\hat{\boldsymbol{p}}$ at each time step. Unfortunately, sampling directly from $\hat{\boldsymbol{p}}$, i.e., ancestral sampling, often leads to quality issues with unnatural, low-probability tokens. Truncation sampling aims to solve this issue post-hoc by choosing a subset of the vocabulary to sample from, setting

---

[1]Code for experiments: https://github.com/mattf1n/basis-aware-threshold.

[2]In the case of natural language, it is not entirely clear what the "true" distribution $\boldsymbol{p}^*$ means exactly. Nonetheless we can use the distribution from which internet text is implicitly sampled as a useful surrogate.

all other tokens to have zero probability. Meister et al. (2023a) frame this strategy as reprioritizing precision over recall (i.e., removing some valid text from the distribution to avoid sampling unlikely text.) We focus on a class of truncation methods that select tokens by choosing a threshold at each timestep and truncating tokens with probability less than that threshold. This simple heuristic has been found to be effective and forms the basis of popular methods like nucleus (top-$p$) (Holtzman et al., 2020) and top-$k$ (Fan et al., 2018) sampling.

Prior work has introduced several heuristics for choosing truncation thresholds. For instance, the threshold can be fixed constant as in $\epsilon$ sampling, or chosen dynamically across different distributions, as in $\eta$, nucleus, top-$k$, and Mirostat sampling (Basu et al., 2021).[3] $\eta$ sampling introduces the idea that the threshold should depend on the entropy of the distribution $H(\hat{p})$ and sets the threshold[4] to $\min(\eta, \sqrt{\eta}H(\hat{p}))$. In the latter three, the threshold is chosen implicitly rather than explicitly, for instance, in nucleus sampling with parameter $\pi$, the threshold is $\min\left\{\hat{p}_i \mid \sum_{\hat{p}_j \geq \hat{p}_i} \hat{p}_j \leq \pi\right\}$.

In the extreme case, truncating all but the most likely token results in greedy decoding. Though this strategy makes it unlikely to sample a token outside the true support, it often results in degenerative patterns like repetition (Holtzman et al., 2020). Furthermore, even for modern language models that suffer less from greedy decoding traps, non-deterministic sample-based decoding is useful for generating multiple completions and for more "creative" generations. Thus, the best choice of threshold must strike a balance between diversity (i.e., including as many tokens as possible in the set of candidates) and coherence (i.e., avoiding sampling tokens outside the true support).

**The softmax bottleneck** The sources of the probability overestimation errors are likely many, but one source of error is particularly compelling and well defined mathematically: the softmax bottleneck (Yang et al., 2018). The softmax bottleneck refers to the limited expressivity of models with a small hidden size and large vocabulary. Recalling the notation from Yang et al. (2018), let $\boldsymbol{A} \in \mathbb{R}^{v \times n}$ be the matrix where each entry $A_{i,j} = \log p^*(i \mid j)$ is the true log-probability of token $i$ given a prefix $j$ from some set of $n > v$ prefixes. Also, let $\boldsymbol{W} \in \mathbb{R}^{v \times d}$ be the softmax matrix for a model, and $\boldsymbol{H} \in \mathbb{R}^{d \times n}$ be the matrix of model hidden states given each prefix. Finally, let $\boldsymbol{J} \in \mathbb{R}^{v \times n}$ be the all-ones matrix. The rank of the model's log-probability matrix

$$\boldsymbol{A}' = \log \operatorname{softmax}(\boldsymbol{W}\boldsymbol{H}) = \boldsymbol{W}\boldsymbol{H} - \boldsymbol{J}\operatorname{diag}(\log \sum_{i=1}^{v} \exp(\boldsymbol{W}\boldsymbol{H})_i) \tag{1}$$

is at most $d+1$ because $\boldsymbol{W}\boldsymbol{H}$ has inner dimension $d$ and therefore rank at most $d$, and the subtrahend has identical rows and therefore has rank at most 1. The rank of $\boldsymbol{A}$ is at most $v$. If the rank of $A$ is much larger than $d$, then $A'$ can be at best a low-rank approximation of $A$. From the Eckart–Young–Mirsk (EYM) theorem for low-rank approximations,

$$\min_{\boldsymbol{A}':\operatorname{rank}(\boldsymbol{A}') \leq d+1} \|\boldsymbol{A} - \boldsymbol{A}'\|_F^2 = \sum_{i=d+2}^{v} \sigma_i^2 \tag{2}$$

where $\|\cdot\|_F$ denotes the Frobenius norm, and $\boldsymbol{\sigma}$ is the vector of singular values of $\boldsymbol{A}$, ordered by decreasing size. Thus, there will always be some error in the model's log-probability estimations if there are more than $d+1$ linearly independent columns in $\boldsymbol{A}$. Yang et al. (2018) hypothesize that this is indeed the case.

Despite these theoretical shortcomings, language models still seem to perform quite well. We hypothesize that the reason for this is that default truncation sampling is sufficient to approximately mitigate errors from the softmax bottleneck. For a deeper discussion, see Appendix A.

## 3 A THEORETICAL EXPLANATION OF TRUNCATION SAMPLING

Given some textual context as input, let $\boldsymbol{p}^*$ denote the true next-token distribution of the language and $\hat{\boldsymbol{p}}$ the model's predicted next-token distribution. Intuitively, if the model's probability *overestimation*

---

[3]Locally typical sampling (Meister et al., 2023b) truncates tokens whose log-probability diverges from the LM's conditional entropy. This may truncate top-probability tokens which likely have non-zero true probability.
[4]Hewitt et al. (2022) instead set $\eta = \min(\varepsilon, \sqrt{\varepsilon}H(\hat{p}))$ for a parameter $\varepsilon$. We diverge for simplicity.

could be additively upper bounded, i.e., if we could show that $\hat{p}_i \leq p_i^* + \tau$ for every token $i$, then this would yield a natural way to avoid sampling tokens not in the support of $p^*$: only sample tokens $i$ with $\hat{p}_i > \tau$ (which, along with the bound, would imply $p_i^* > 0$). This is exactly what truncation sampling does. However, a difficulty in motivating truncation sampling via this argument is that it is unclear how to derive such an additive upper bound on probability overestimation.

Our key observation is that $\mathbf{A}'$ being a low-rank approximation of $\mathbf{A}$ can be used to conclude that the model's log-probability *underestimation* is non-zero but additively upper bounded. Indeed, assuming $\mathbf{A}'$ is a reasonably good low-rank approximation of $\mathbf{A}$, Equation 1 implies such an upper bound in the log-probability space, which yields a multiplicative upper bound in the probability space. We then combine this underestimation upper bound with basic properties of a probability distribution in order to derive the desired *additive* upper bound on the model's probability *overestimation*. Lastly, we show formally how this overestimation upper bound directly motivates truncation sampling.

## 3.1 BOUNDING LOG-PROBABILITY UNDERESTIMATION

We begin by proving bounds on models' log-probability errors. Specifically, we find bounds on the maximum log-probability underestimation error of the model, $\max(\mathbf{A} - \mathbf{A}')$. We focus exclusively on underestimation errors because log-probability overestimation errors cannot be bounded above.[5]

**Maximum log-probability error upper bound**  We begin by upper-bounding all model's log-probability underestimations. In particular, the underestimation errors $\mathbf{A} - \mathbf{A}'$ are upper-bouded by $\max(\mathbf{A} - \mathbf{A}') \leq \max \mathbf{A} - \min \mathbf{A}' \leq - \min \mathbf{A}'$, where the last inequality holds because $\max \mathbf{A}$ is a log-probability and hence upper-bounded by $0$. In other words, the negative minimum log-probability prediction $\min \mathbf{A}'$ upper bounds all underestimation. As an example, a uniform predicted distribution underestimates the log-probability of a token by at most $-\log(1/v)$.

**Maximum log-probability error lower bound**  Next, we lower-bound maximum underestimation errors by showing that they are strictly positive. We conjecture that this lower-bound on error is loose, i.e., that the maximum error is bounded away from $0$, depending on the singular values of $\mathbf{A}$.

## 3.2 BOUNDING PROBABILITY OVERESTIMATION

Having established bounds on maximum *log-probability underestimation*, we now show that assuming such an upper bound implies an additive upper bound on maximum *probability overestimation*. As before, fix some input textual context and let $\boldsymbol{p}^*$ and $\hat{\boldsymbol{p}}$ denote the true and model's predicted next-token distributions, respectively, for that context.

**Theorem 1.** *If $\log \hat{p}_i$ underestimates $\log p_i^*$ by at most $\delta$ for all tokens $i$, then $\hat{p}_i$ overestimates $p_i^*$ by at most $1 - \exp(-\delta)$ for all tokens $i$.*

See Appendix D for a proof. Note that the precondition $\log p_i^* - \log \hat{p}_i \leq \delta$ implies $\hat{p}_i \geq p_i^* \exp(-\delta)$. Intuitively, since $\hat{p}$ is a valid probability distribution summing to $1$, if it cannot underestimate token probabilities beyond a factor of $\exp(-\delta)$, then it also cannot overestimate other tokens' probabilities beyond a certain additive factor. We compute this additive factor and find it to be $1 - \exp(-\delta)$.

## 3.3 EXPLAINING TRUNCATION SAMPLING

Recall that threshold-based truncation sampling works by only sampling tokens with probability greater than some threshold $\tau$. Sampling methods that choose a different $\tau$ at every time step can be viewed as additional heuristics for guessing when model outputs will have smaller errors. Theorem 1 provides a direct explanation for why threshold-based truncation sampling might be successful:

**Corollary 1** (Threshold-based truncation works)**.** *Suppose $\log \hat{\boldsymbol{p}}$ underestimates $\log \boldsymbol{p}^*$ by at most $\delta$. Then, for any threshold $\tau \geq 1 - \exp(-\delta)$, threshold-based truncation sampling discards all tokens that are not in the support of $\boldsymbol{p}^*$.*

---

[5]If the true distribution assigns zero probability to some tokens in some contexts (e.g., $p^*(\text{"ate"} \mid \text{"I went to the"}) = 0$), then the corresponding log-probability is $-\infty$. Hence any finite log-probability estimate will have infinite error.

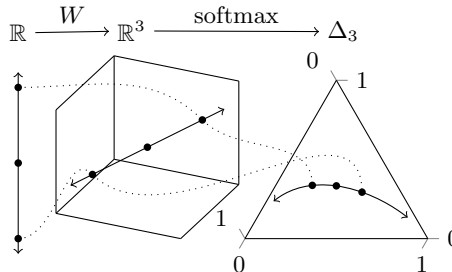

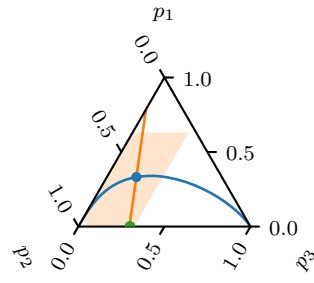

Figure 2: For a toy model with hidden size 1, vocabulary size 3, and an embedding matrix $\boldsymbol{W} \in \mathbb{R}^{3 \times 1}$, $\boldsymbol{W}$ projects the space of possible hidden states $\mathbb{R}$ into a 1-dimensional subspace of the space of possible logits $\mathbb{R}^3$. In turn, the softmax function projects this 1D logit subspace onto a 1D subspace of the space $\Delta_3$ of possible probability distributions over 3 tokens. Thus, our toy model can only output distributions within a 1D subspace of $\Delta_3$, which is the image of $\mathrm{softmax} \circ \boldsymbol{W}$.

Figure 3: If the model outputs $\hat{\boldsymbol{p}}$ (the blue dot) within the space of possible outputs (blue line), then each token $i$ might have zero true probability only if there is a distribution $\boldsymbol{p}$ with $p_i = 0$ that satisfies both the BA constraints (orange line) and the threshold constraints (orange area). For example, the orange line and area coincide at the green dot where $p_1 = 0$, therefore token 1 might have zero true probability. The other tokens must have nonzero true probability since there are no other such solutions.

Furthermore, based on the above proof, we present an alternative formulation of truncation sampling.

**Corollary 2** (Threshold sampling reformulation). *For a model with maximum log-probability underestimation error $\delta$, if the model outputs $\hat{\boldsymbol{p}}$ and there is no distribution $\boldsymbol{p}$ with $p_i = 0$ such that $p_j \leq \hat{p}_j \exp(\delta)$ for $j \in \{1, 2, \ldots, v\}$, then $p_i^* > 0$.*

This follows directly from Equation (8) from the proof in the appendix, and is the contrapositive of the more straightforward statement that if $p_i^* = 0$ then there exists a distribution satisfying inequality conditions in the corollary, namely $\boldsymbol{p}^*$. One can check that only sampling tokens based on Corollary 2 yields the same candidate sets as threshold sampling with $1 - \exp(-\delta)$ as the parameter. This alternative formulation will become useful later on when we combine methods for proving certain tokens are in the support.

## 4 DIRECTLY ADDRESSING ERRORS FROM THE SOFTMAX BOTTLENECK

The previous section demonstrates that we can arrive at truncation sampling by making an assumption about the log-probability errors, which allows us to prove that certain tokens have true probability greater than zero. However, truncating via a threshold is an inherently limited approach: if a model assigns more probability to a "bad" (zero true probability) token than a "good" (nonzero true probability) token, then there is no threshold that discards the bad token without discarding the good token. Naïvely, it would seem that this type of issue is unsolvable, however, it turns out that if this error was is caused by the softmax bottleneck, we can actually recover the good token without risking sampling the bad token. By exploiting $\boldsymbol{W}$, the low-rank basis for the model's outputs, and we can deduce exactly which tokens may have errors due to the softmax bottleneck, regardless of their relative probability. In this section we show mathematically how we can extend threshold sampling to take full advantage of our knowledge of the softmax bottleneck.

### 4.1 BASIS-AWARE SAMPLING

At a high level, we will motivate this approach by showing that the function used to transform the hidden state $\boldsymbol{h}$ to a probability distribution $\hat{\boldsymbol{p}}$ restricts model's outputs to a subset of the possible probability distributions. When the true distribution $\boldsymbol{p}^*$ lies outside of this set, then we can expect the model to output the $\hat{\boldsymbol{p}}$ within the set that minimizes the model's training loss with respect to $\boldsymbol{p}^*$. We can exploit this property to identify the set of distributions wherein the true distribution lies,

namely the set of distributions that $\hat{\boldsymbol{p}}$ minimizes loss with. If no distributions within this set assign zero probability to a particular token, then that token must have nonzero probability.

To build intuition for how a model's outputs are restricted, consider the toy model in Figure 2. We generalize this toy model to a model with hidden size $d$ and vocabulary size $v$. Observe that the composed functions $\mathrm{softmax} \circ \boldsymbol{W}$ define a linear map: first, the model's softmax matrix $\boldsymbol{W} \in \mathbb{R}^{v \times d}$ defines a linear map $\mathbb{R}^d \to \mathbb{R}^v$. Next, it is a lesser-known fact that the softmax function is a linear map from $\mathbb{R}^v \to \Delta_v$, where $\Delta_v$ is the $(v-1)$-dimensional vector space of valid probability distributions over $v$ variables (Aitchison, 1982) (see Appendix C for an explanation). Therefore, $\mathrm{softmax} \circ \boldsymbol{W} : \mathbb{R}^d \to \Delta_v$ is a linear map from a $d$-dimensional space to a $(v-1)$-dimensional space, meaning the image of this function is an at-most $d$-dimensional subspace of $\Delta_v$. In other words, the space of model outputs is restricted to a subset of all possible probability distributions over the vocabulary.[6]

What distribution should a model output, given that the true distribution $\boldsymbol{p}^*$ may not lie in the subspace of possible outputs? Typically, language models are trained to minimize cross-entropy with the true distribution. Therefore, a well-trained model can be expected to output the distribution $\hat{\boldsymbol{p}}$ within the image of $\mathrm{softmax} \circ \boldsymbol{W}$ that minimizes cross-entropy with $\boldsymbol{p}^*$. In other words, we assume that the model will produce the hidden state $\boldsymbol{h}$ such that $\mathrm{crossentropy}(\mathrm{softmax}(\boldsymbol{W}\boldsymbol{h}), \boldsymbol{p}^*)$ is minimized. The key insight of our method is that if $\boldsymbol{h}$ does not minimize cross entropy with *any* distribution $\boldsymbol{p}$ such that $p_i = 0$, then $p_i^* \neq 0$, i.e., token $i$ is in the true support.

**Theorem 2** (Basis-aware sampling). *If $\hat{\boldsymbol{p}}$ is the predicted distribution from a cross-entropy-minimizing model with embedding matrix $\boldsymbol{W}$, and if there is no valid probability distribution $\boldsymbol{p}$ such that $p_i = 0$ and $\boldsymbol{W}^\top \boldsymbol{p} = \boldsymbol{W}^\top \hat{\boldsymbol{p}}$, then the token's true probability $p_i^*$ is greater than $0$.*

See proof in Appendix D. This gives us a new way to prove that tokens are in the true support, similar to Corollary 2, but in a way that directly compensates for errors due to the softmax bottleneck.

## 4.2 COMBINING SAMPLING METHODS

Theorem 2 and Corollary 2 equip us with methods for proving tokens are in the true support. By combining the constraints specified from each method we can create a hybrid proof strategy to take advantage of both methods' insights. In particular, if there does not exist a distribution $\boldsymbol{p}$ with $p_i = 0$ such that $p_j \leq \hat{p}_j \exp(\delta)$ for all $j$ (the truncation constraint) *and* $\boldsymbol{W}^\top \boldsymbol{p} = \boldsymbol{W}^\top \hat{\boldsymbol{p}}$ (the basis-aware constraint), then $p_i^* > 0$.

This hybrid proof strategy naturally yields a sampling method: sample only tokens that we can prove are in the support. We call this method *basis-aware threshold* (BAT) sampling. Fortunately, both the threshold constraint and basis-aware (BA) constraints are linear, so we can use an off-the-shelf linear programming optimizer to verify whether a token is in the support. Concretely, if the optimizer determines that there does not exist a feasible solution $\boldsymbol{p} \in \mathbb{R}^v$ such that:

$$ p_i = 0, \quad \sum_{j=1}^{v} p_j = 1, \quad \forall j : 0 \leq p_j \leq \hat{p}_j \exp(\delta), \quad \boldsymbol{W}^\top \boldsymbol{p} = \boldsymbol{W}^\top \hat{\boldsymbol{p}}, \tag{3} $$

then $p_i^* > 0$. Thus, our sampling strategy can be: sample a token $i$ according to the model's output probabilities; if the optimizer finds a solution to (3), reject the token and re-sample; otherwise accept. See Algorithm 1 in the Appendix.

We expose $\delta$ as a parameter to tune the restrictiveness of the sampling method. For large $\delta$, BAT becomes more like greedy sampling, and for small $\delta$, more like ancestral sampling. The value of $\delta$ can be chosen on a per-context basis using any threshold sampling heuristic, be it $\epsilon$, $\eta$, or nucleus sampling. Given a threshold $\tau$ from the heuristic, set $\exp \delta = 1/(1-\tau)$. We call these variants of BAT sampling BA-$\epsilon$, BA-$\eta$, an BA-nucleus sampling.

**A toy example** Suppose our model has hidden size 1, vocabulary size 3, and embedding matrix $\boldsymbol{W}^\top = [0.55 \quad 0.71 \quad 0.29]$. We employ the truncation sampling assumption that our model's output distributions are somewhat close to the true distribution by saying $p_i^* \leq \hat{p}_i \exp \delta$ and choosing

---

[6]may correctly observe that the "bottleneck" is a consequence of the linear map $\boldsymbol{W}$, not the softmax function. We keep our notation for the sake of consistency with Yang et al. (2018).

$\delta = \log 1.9$ so that $p_i^* \leq 1.9\hat{p}_i$ for all tokens $i$. Additionally, assume the model's outputs minimize cross-entropy with the true distribution, i.e., $\boldsymbol{W}^\top \boldsymbol{p}^* = \boldsymbol{W}^\top \hat{\boldsymbol{p}}$ for all $\hat{\boldsymbol{p}}$. Now suppose our model outputs $\boldsymbol{h} = [2.55]$. The output distribution is therefore $\hat{\boldsymbol{p}} = \text{softmax}(\boldsymbol{Wh}) = [0.33 \quad 0.50 \quad 0.17]^\top$.

Our strategy only samples tokens for which we can prove that the true probability is positive. Referring to Figure 3, we see that there are no probability distributions $\boldsymbol{p}$ that satisfy our assumptions with $p_2 = 0$ or $p_3 = 0$. However, $\boldsymbol{p} = [0 \quad 0.70 \quad 0.30]$ *does* satisfy our assumptions. Therefore, if we sample token 1 we should reject it, as we only have evidence that $p_2^* \neq 0$ and $p_3^* \neq 0$. Notice that this strategy is non-monotonic: $\hat{p}_1 > \hat{p}_3$, but we only reject token 1, not token 3.

**Basis-aware threshold sampling in practice**    The proposed implementation of basis-aware sampling requires solving rather large linear programs, which tends to be too computationally expensive to be practical, even when using proprietary solvers. The long run times can mainly be attributed to the size of $\boldsymbol{W}$. To make BAT feasible in practice, we approximate the full solution by replacing $\boldsymbol{W}$ with an much smaller matrix such that no additional tokens are accepted, and the set of rejected tokens minimally increases. More details are deferred to Appendix E. This shortens the run time from over a minute on a proprietary solver to about a second. We further reduce the generation run time by observing that whenever a token has probability greater than $1 - \exp(-\delta)$ we can safely accept it without running the program, since the program will be infeasible. Since high-probability tokens are most likely to be sampled, the program only needs to run once every few samples. The amortized cost of BAT sampling comes to only about 0.1 seconds per token as the program typically runs every 10 samples.

## 5   PILOT EXPERIMENTS WITH BASIS-AWARE TRUNCATION

We conduct several evaluations with GPT-2 to pilot BAT sampling as a viable alternative to threshold sampling. While more powerful language models exist, these models suffice since we are primarily interested in testing the effect of the BAT sampling on performance under controlled settings.

As baseline methods for comparison, we select $\eta$, $\epsilon$, and nucleus sampling (see §2). We also use $\eta$ and $\epsilon$ as methods for selecting the $\delta$ parameter at each time step for BAT sampling. In preliminary experiments, we also tried BA-nucleus, but found it to be significantly worse. One possible intuition for why is that the methods for choosing the threshold $\epsilon$ and $\eta$ are similar to the formulation of threshold sampling used to develop BAT. Nucleus sampling on the other hand determines the threshold using a function that is somewhat inconsistent with our framework.

We evaluate models on open-ended generation using both human annotators and automatic metrics. For each model and sampling setting, we generate completions for 5000 35-token prefixes taken from the Open Web Text (OWT) (Gokaslan et al., 2019). We use OWT because it comes from a similar distribution to GPT-2's training data. We report MAUVE (Pillutla et al., 2021) similarity between human text and generated text for parameter selection and automatic evaluation.

**Parameter selection and evaluation**    We perform a parameter sweep for nucleus, $\eta$, and $\epsilon$ sampling and select the parameter that gives the highest MAUVE score on the OWT validation set (see Table 3 in the appendix). We control for the parameter choice in comparisons between BAT methods and their vanilla counterparts, by matching the parameters by selecting the BAT parameter that rejects the same proportion of tokens from corpus of human text as the vanilla method; see Appendix F for more details. Using these parameters, we generate completions on the OWT test set for automatic evaluation with MAUVE and human evaluation.

### 5.1   QUALITATIVE, AUTOMATIC, AND HUMAN EVALUATION

**Qualitative analysis**    Figure 4 shows the effects of truncation methods on the next-token distributions from 6 prefixes, drawn from Hewitt et al. (2022). Unlike threshold sampling methods, BAT can reject low-quality high-probability tokens while accepting high-quality low-probability tokens.

**BA-$\eta$ outperforms all other methods for GPT-2-Large**    We compare the MAUVE scores on OWT for each method and model size in Figure 5. The results show that no single method consistently performs best, with BAT methods sometimes out-performing and sometimes under-performing their

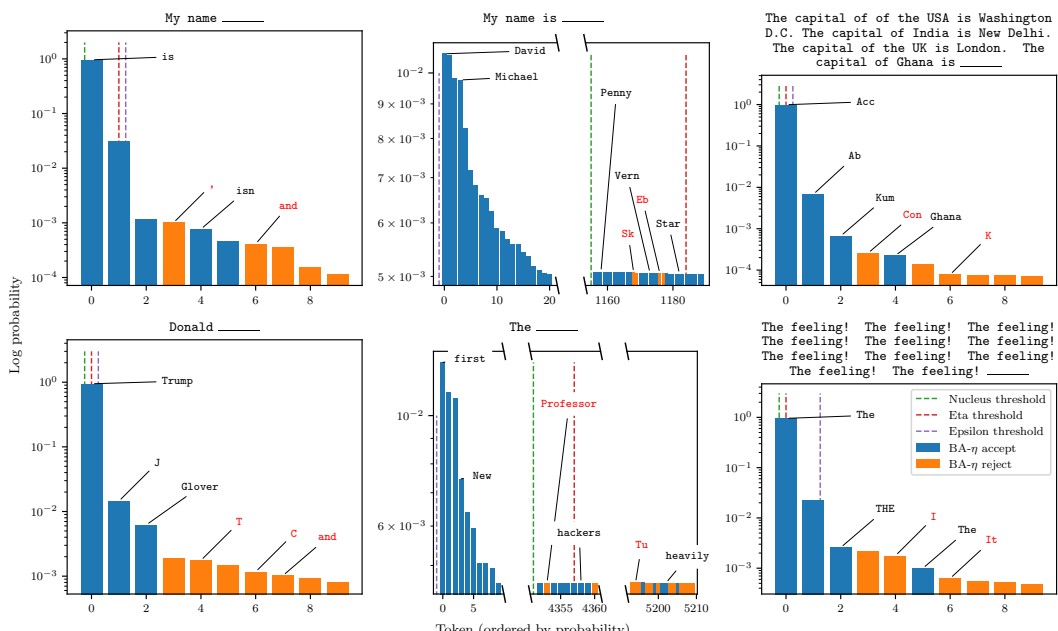

Figure 4: Additional qualitative examples, following the same setup as Figure 1.

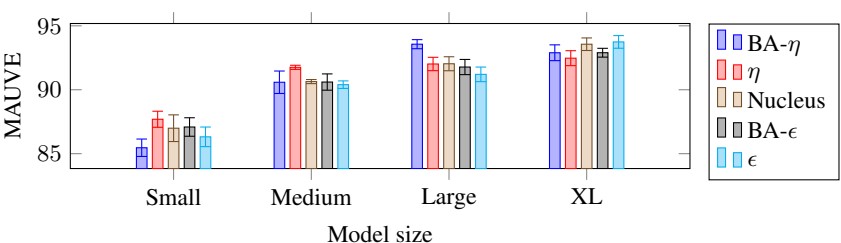

Figure 5: MAUVE scores for sampling methods on Open Web Text test set. No single sampling method consistently outperforms across sizes. BA-$\eta$ performs remarkably well for GPT-2-Large.

vanilla counterparts. We do, however, see that BA-$\eta$ outperforms $\eta$ sampling for the two larger model sizes, and does particularly well against all methods for GPT-2-Large.

**BA-$\eta$ outperforms $\eta$ sampling in low-entropy decoding across model sizes**  We compare BA-$\eta$ and $\eta$ sampling across different $\eta$ parameters, again matching our BA-$\eta$ parameter to reject the same proportion of human text as the $\eta$ parameter. As shown in Figure 6, we find that for more restrictive sampling (i.e., larger $\eta$, closer to greedy decoding), BA-$\eta$ consistently outperforms $\eta$ sampling. To verify our results (since we know from Figure 5 that model size effects which method is best) we show in Table 1 that this pattern holds across all model sizes.

**More constraints improves BAT**  Since we reduce the number of constraints in the linear program to make it run quickly, we can add constraints back into to program to verify that the basis-aware constraints are the reason for the gains in BAT sampling. We again adjust the BAT parameter to match the proportion of rejected human text to control for the additional tokens added to the support from the new constraints. Figure 7 shows that adding more BA constraints indeed increases the MAUVE score for our method. This is direct evidence that controlling for the softmax bottleneck helps reduce errors in the model distribution.

**Human annotators narrowly favor BA-$\eta$ and prefer coherence to diversity**  To support our automatic evaluations, we additionally use human annotators from Amazon Mechanical Turk to compare both methods. Annotators are tasked with pairwise comparisons between generations from

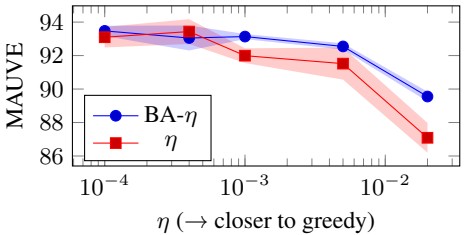

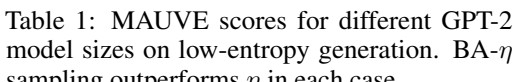

$\eta \ (\rightarrow \text{closer to greedy})$

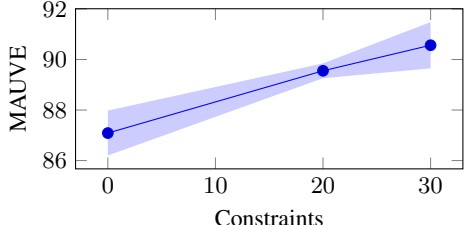

Constraints

Figure 6: MAUVE scores for GPT-2-XL with BA-$\eta$ and $\eta$ sampling for different $\eta$. Under low-entropy generation (i.e., closer to greedy), BA-$\eta$ consistently outperforms $\eta$ sampling.

Figure 7: MAUVE scores for GPT-2-XL as the number of BA constraints varies. BAT sampling improves with more constraints.

Table 1: MAUVE scores for different GPT-2 model sizes on low-entropy generation. BA-$\eta$ sampling outperforms $\eta$ in each case.

Table 2: Pairwise human evaluation results. BA-$\eta \equiv x$ indicates the BA-$\eta$ parameter chosen to match $\eta = x$.

| Size Method | Small | Medium | Large | XL |
|---|---|---|---|---|
| $\eta$ | $85.0_{1.4}$ | $90.4_{0.1}$ | $86.0_{0.5}$ | $87.1_{1.2}$ |
| BA-$\eta$ | $\mathbf{87.8_{1.0}}$ | $\mathbf{92.2_{0.6}}$ | $\mathbf{88.4_{0.5}}$ | $\mathbf{89.6_{0.4}}$ |

| Method 1 | Method 2 | 1 wins | 2 wins | Tie |
|---|---|---|---|---|
| BA-$\eta \equiv 0.002$ | $\eta = 0.002$ | 0.43 | 0.38 | 0.19 |
| BA-$\eta \equiv 0.024$ | $\eta = 0.024$ | 0.48 | 0.47 | 0.05 |
| BA-$\eta \equiv 0.024$ | BA-$\eta \equiv 0.001$ | 0.50 | 0.42 | 0.08 |

each method and generated from the same prefix. See Appendix F.1 for more details. Table 2 shows that, annotators narrowly prefer generations from BA-$\eta$ sampling to those from $\eta$ sampling. Furthermore we see that human annotators prefer lower entropy generations. This is likely because humans only see 1 generation per method, making it impossible to assess diversity in the generations.

## 5.2 DISCUSSION

Overall, our results provide empirical evidence that the softmax bottleneck is responsible for significant errors in language model next-token distributions, and show that BAT sampling offers a viable method for mitigating those errors. Under low-entropy generation, BAT offers clear advantages to threshold sampling, where only a few tokens are permissible.

Although our pilot study shows promising results for BA-$\eta$ sampling in low-entropy generation settings, there remain a number of limitations. For instance, as mentioned in §5, BAT does not pair well with nucleus sampling. Furthermore, we find that for certain prefixes and sufficiently low-entropy sampling parameters, BA-$\epsilon$ accepts no tokens. This is a non-issue for threshold sampling which can fall back to greedy sampling, but because BAT relies on rejection sampling, it is not known when to revert to greedy. Though it is possible to implement a max-retries guard, this remains computationally expensive and the generations themselves tend to degrade.

A broader issue that BAT must deal with is the expensive computation associated with running the linear program. While this is generally not an issue for generation, certain tasks are infeasible, such as finding the exact set of candidate tokens, which would require running the linear program on the full vocabulary. We remain optimistic that further optimizations to the method can be made to allow this in future work, as well as enable BAT sampling with higher constraint counts.

## 6 CONCLUSION

Our work fills a crucial gap in the theoretical understanding of truncation sampling methods and how they account for language model errors. These theoretical findings translate into a more direct method for mitigating errors due to the softmax bottleneck. As a result, our BAT sampling method can discard higher-probability tokens while keeping higher-quality but lower-probability tokens. Lastly, our pilot study with BAT sampling shows promising results in low-entropy generation.

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

## A  FURTHER RELATED WORK

**Generating from autoregressive language distributions.** Generating strings from autoregressive, optionally conditional, generative models of language has a long history in NLP; for decades, algorithms were developed for approximating the maximum-likelihood string under the model (Jelinek, 1990), e.g., beam search (Reddy, 1977), under the understanding that there is one best output for, e.g., speech recognition. As Transformers became used for general-purpose, and often high-entropy generation wherein one wants to be able to generate multiple good completions, it was found that search for the highest-likelihood strings led to low-quality generations (Fan et al., 2018; Holtzman et al., 2020; Hashimoto et al., 2019; Stahlberg & Byrne, 2019; Meister et al., 2020). In developing algorithms for high-entropy generation, the afore-mentioned line of work attempts to maintain the learned distribution as much as possible (Holtzman et al., 2020; Hewitt et al., 2022); another significant design principle has related to the *uniform information density* principle that humans are observed to obey, motivating Meister et al. (2023b). This algorithm intentionally deviates from the overall distribution more, by sometimes truncating high-probability tokens in order to never generate any tokens that are *too* high probability relative to the overall entropy. Krishna et al. (2022) show that language models do not effectively make use of long-term context, finding that an explicitly trained re-ranker can help. Li et al. (2023) hypothesizes that language model errors are distributed similarly in small models as in large, showing that taking the *difference* of their logits can help improve the large model's generations. Some ideas from high-entropy generation, including this, and the $\epsilon$-sampling algorithm, have shown to be useful even in low-entropy generation where previously algorithms like beam search have performed best (Freitag et al., 2023; Sennrich et al., 2023).

**Did the softmax bottleneck turn out not to be a problem?** After the demonstration of the softmax bottleneck by Yang et al. (2018), various algorithms were proposed for efficiently learning a high-rank language models (Yang et al., 2019; Ganea et al., 2019). Chang & McCallum (2022) showed that the softmax bottleneck makes certain multi-mode distributions difficult to model, while Demeter et al. (2020) demonstrated that the low-rank nature of language models means that it is possible for certain word tokens to be *unable* to be the argmax, but Grivas et al. (2022) demonstrated that this is rarely the case in practice. Overall, rank considerations have not been at the fore of language model development, as language models have scaled, their hidden state sizes have scaled as well, but stayed smaller than their vocabulary sizes (Scao et al., 2022; Biderman et al., 2023; Touvron et al., 2023). Throughout this time, when one *generates* from language models, one almost always lowers entropy and performs some kind of truncation sampling (or in the extreme, greedy decoding). Our results suggest that training high-rank language models may appear unnecessary because default truncation sampling mitigates errors stemming from the low-rank approximation.

## B  BAT SAMPLING ALGORITHM

Algorithm 1 gives the procedure for BAT sampling.

---

**Algorithm 1** BAT sampling

---
1: **procedure** BAT(Threshold $\tau$, Next-token distribution $\hat{\boldsymbol{p}}$)
2:     **repeat**
3:         Sample $i \sim \hat{\boldsymbol{p}}$
4:     **until** $\nexists \boldsymbol{p}$ such that $p_i = 0 \wedge \boldsymbol{W}^\top \boldsymbol{p} = \boldsymbol{W}^\top \hat{\boldsymbol{p}} \wedge \forall j, p_j \leq \hat{p}_j/(1-\tau)$        ▷ Use LP solver
5:     **return** $i$
6: **end procedure**

---

## C  THE SOFTMAX FUNCTION IS LINEAR

It is an unintuitive fact that the softmax function is a linear map $\mathbb{R}^d \to \Delta_d$. The key here is that addition and scalar multiplication are defined on $\Delta_d$ in a non-standard way. The elements of $\Delta_d$ are tuples of length $d$ whose entries sum to one. Vector addition $\oplus$ in $\Delta_d$ is defined as elementwise multiplication followed by normalization

$$\boldsymbol{p} \oplus \boldsymbol{q} = \frac{\boldsymbol{p} \odot \boldsymbol{q}}{\sum_{i=1}^d p_i q_i}, \tag{4}$$

and multiplication $\otimes$ by a constant $\lambda \in \mathbb{R}$ is elementwise exponentiation followed by normalization

$$\lambda \otimes \boldsymbol{p} = \frac{\boldsymbol{p}^\lambda}{\sum_{i=1}^d p_i^\lambda}. \tag{5}$$

One can check that these operations satisfy the axioms of a vector space, and that the softmax function satisfies additivity and homogeneity under these operations, i.e.,

$$\mathrm{softmax}(\boldsymbol{u} + \boldsymbol{v}) = \mathrm{softmax}(\boldsymbol{u}) \oplus \mathrm{softmax}(\boldsymbol{v}) \tag{6}$$

and

$$\mathrm{softmax}(\lambda \boldsymbol{u}) = \lambda \otimes \mathrm{softmax}(\boldsymbol{u}). \tag{7}$$

## D  PROOFS

*Proof of Theorem 1.* By the precondition of the theorem, we have $\log p_i^* - \log \hat{p}_i \leq \delta$ for all $i$. It follows that:

$$\hat{p}_i \geq p_i^* \exp(-\delta). \tag{8}$$

Intuitively, since $\hat{p}$ is a valid probability distribution summing to 1, if it cannot underestimate token probabilities beyond a factor of $\exp(-\delta)$, then it also cannot overestimate other tokens' probabilities beyond a certain factor; we will show that this factor is $1 - \exp(-\delta)$.

To this end, we consider each token individually and calculate the maximum possible probability overestimation based on the maximum probability underestimation of the other tokens. Keeping in mind that any probability added to a token must be removed from other tokens to preserve a valid probability distribution, the maximum probability added to a token is the sum of the maximum probabilities subtracted from the other tokens. This gives us that for all $i$:

$$\hat{p}_i - p_i^* = \sum_{k \neq i} p_k^* - \sum_{k \neq i} \hat{p}_k \tag{9}$$

$$\leq \sum_{k \neq i} \left( p_k^* - p_k^* \exp(-\delta) \right) \qquad \text{From (8)} \tag{10}$$

$$= (1 - \exp(-\delta)) \sum_{k \neq i} p_k^* \qquad \text{Factor out } p_k^* \tag{11}$$

$$= (1 - \exp(-\delta))(1 - p_i^*) \qquad \text{Probabilities sum to 1} \tag{12}$$

$$\leq 1 - \exp(-\delta) \qquad\qquad 0 \leq p_i^* \leq 1. \tag{13}$$

We thus have our desired probability overestimation bound, starting with the assumption of a log-probability underestimation bound. $\qquad\square$

*Proof of Theorem 2.* We begin by assuming that our model has learned to minimize cross-entropy with the true distribution, implying that

$$\frac{\partial}{\partial \boldsymbol{h}} \text{crossentropy}(\text{softmax}(\boldsymbol{W}\boldsymbol{h}, \boldsymbol{p}^*)) = 0. \tag{14}$$

Expanding and simplifying this equation, we can obtain

$$\frac{\partial}{\partial \boldsymbol{h}} \left( -\sum_i p_i^* \log(\text{softmax}(Wh)_i) \right) = 0 \qquad \text{cross entropy defn.} \tag{15}$$

$$\frac{\partial}{\partial \boldsymbol{h}} \left( -\sum_i p_i^* Wh_i - p_i^* \log \sum_j \exp(Wh)_j \right) = 0 \qquad \text{Log of softmax} \tag{16}$$

$$\frac{\partial}{\partial \boldsymbol{h}} \sum_i p_i^* \log \sum_j \exp(Wh)_j = \frac{\partial}{\partial \boldsymbol{h}} \sum_i p_i^* (Wh)_i \qquad \text{Distribute } \frac{\partial}{\partial h} \tag{17}$$

$$\frac{\partial}{\partial \boldsymbol{h}} \log \sum_j \exp(Wh)_j = \frac{\partial}{\partial \boldsymbol{h}} \sum_i p_i^* (Wh)_i \qquad \sum_i p_i^* = 1 \tag{18}$$

$$\frac{\sum_j \exp(Wh)_j \frac{\partial}{\partial \boldsymbol{h}}(Wh)_j}{\sum_j \exp(Wh)_j} = \frac{\partial}{\partial \boldsymbol{h}} \sum_i p_i^* (Wh)_i \qquad \text{Derivative} \tag{19}$$

$$\frac{\partial}{\partial \boldsymbol{h}} (\boldsymbol{W}\boldsymbol{h})^T \frac{\exp(\boldsymbol{W}\boldsymbol{h})}{\sum_j \exp(\boldsymbol{W}\boldsymbol{h})_j} = \frac{\partial}{\partial \boldsymbol{h}} (\boldsymbol{W}\boldsymbol{h})^T \boldsymbol{p}^* \qquad \text{Factor} \tag{20}$$

$$\frac{\partial}{\partial \boldsymbol{h}} (\boldsymbol{W}\boldsymbol{h})^T \text{softmax}(\boldsymbol{W}\boldsymbol{h}) = \frac{\partial}{\partial \boldsymbol{h}} (\boldsymbol{W}\boldsymbol{h})^T \boldsymbol{p}^* \qquad \text{Softmax defn.} \tag{21}$$

$$\boldsymbol{W}^T \hat{\boldsymbol{p}} = \boldsymbol{W}^T \boldsymbol{p}^* \qquad \text{Derivative} \tag{22}$$

where $\hat{\boldsymbol{p}}$ is the output distribution of the model. Thus, if there does not exist any valid probability distribution $\boldsymbol{p}$ such that $p_i = 0$ and $\boldsymbol{W}^T \boldsymbol{p} = \boldsymbol{W}^T \hat{\boldsymbol{p}}$, then $p_i^* \neq 0$. □

## E  BASIS-AWARE THRESHOLD SAMPLING IN PRACTICE

Basis-aware sampling presents a number of practical challenges. Chief among them is the sheer size of the linear programs to be solved. These programs have $v$ variables and $d + 2v + 2$ constraints. No open-source solver we tried was able to solve a single problem in a reasonable amount of time, avoid hitting a numerical errors, and solve within its default max-iteration limits. Proprietary solvers do better in some cases, but only the MOSEK solver (ApS, 2023) was able to solve the full problem in under 1 minute. Even this relatively faster solving rate makes text generation at scale impractical.

To address this, we reduce the size of the linear program dramatically by discarding many constraints. While doing so, however, we also aim to maintain as much of the original solution space as possible, so as to minimize the effect on the set of tokens discarded by basis-aware sampling.[7]

In order to reduce the number of constraints originating from the $W^T \boldsymbol{p} = W^T \hat{\boldsymbol{p}}$ term from $d$ to $c$, we can simply discard any $d - c$ columns of $W$ to obtain $W^c$. Clearly, if $\boldsymbol{p}$ satisfies $W^T \boldsymbol{p} = W^T \hat{\boldsymbol{p}}$, it will continue to also satisfy $W^{cT} \boldsymbol{p} = W^{cT} \hat{\boldsymbol{p}}$. Thus, if a token was originally rejected by bottleneck-aware sampling, it would still be rejected, i.e., using $W^c$ instead of $W$ does not add new candidate tokens. It may, however, remove some candidates, and we would like to minimize this effect.

Suppose $W$ has rank $b \leq d$. Then the set of probability distributions $\boldsymbol{p}$ satisfying $W^T \boldsymbol{p} = W^T \hat{\boldsymbol{p}}$ forms a linear subspace $S \subseteq \mathbb{R}^v$ of dimension $d - b$. Further, $W^c$ has rank at most $\min\{b, c\}$, implying the set of distributions $\boldsymbol{p}$ satisfying the relaxed condition $W^{cT} \boldsymbol{p} = W^{cT} \hat{\boldsymbol{p}}$ forms a linear superspace $S^c$ of $S$ of dimension *at least* $d - \min\{b, c\}$. Recall that the larger $S^c$ is, the more candidate tokens will be removed by bottleneck sampling. Thus, to minimize candidate removal, we seek an $S^c$ that is of dimension *exactly* $d - \min\{b, c\}$. This can be achieved easily by keeping in $W^c$ any set of $\min\{b, c\}$ linearly independent columns of $W$. Note that if $b \leq c$, the use of such a $W^c$

---

[7]Without any constraints, basis-aware threshold sampling reduces to basic threshold sampling.

will, in fact, not remove *any* candidate, as $S^c$ will equal $S$. Otherwise $S^c$ will be a $d - c$ dimensional superspace of $S$.

When $b > c$, however, this solution is still not optimal, as which linearly independent columns of $W$ we choose to keep in $W^c$ determines how "close" $S^c$ will be to the original solution space $S$. Intuitively, we would like to preserve $S$ along dimensions that correspond to the $c$ largest eigenvalues of $W$. To accomplish this, we turn to singular value decomposition: find three matrices $U \in \mathbb{R}^{v \times d}$, $\Sigma \in \mathbb{R}^{d \times d}$, and $V \in \mathbb{R}^{d \times d}$ such that $W = U \Sigma V^T$, then replace $W$ with $U^c \in \mathbb{R}^{v \times c}$, where $U^c$ represents the first $c$ columns of $U$. Since $U$ is simply a linear transformation of $W$, the solutions (in terms of $\boldsymbol{p}$) of $U^T \boldsymbol{p} = U^T \hat{\boldsymbol{p}}$ are precisely the subspace $S$ of dimension $d - b$ as before. Again, as before, replacing $W$ with $U^c$ does not add new tokens to the set of candidates, and may remove some candidates when $b > c$. Importantly, when $b > c$, $U^c$ will intuitively be the "closest" possible approximation of $W$ (capturing its $c$ largest eigenvalues). Thus, $S^c$ will form a desirable approximation of $S$.

The above SVD based approximation is what we use in practice. This reduces the number of constraints from $d$ ($\approx$ 700-1200 for our models) to $c$ (typically 20), and shortens the run time from over a minute on a proprietary solver to about a second.

## F  PARAMETER SELECTION

Table 3: Parameter sweeps and chosen parameters for each method and size

| Method | Sweep | Small | Medium | Large | XL |
|---|---|---|---|---|---|
| Nucleus | $\{0.89, 0.9, 0.92, 0.95, 0.99\}$ | 0.92 | 0.89 | 0.92 | 0.95 |
| $\epsilon$ | $\{0.0003, 0.0006, 0.0009, 0.001, 0.002\}$ | 0.0009 | 0.0003 | 0.0009 | 0.0003 |
| $\eta$ | $\{0.0003, 0.0006, 0.0009, 0.002, 0.004\}$ | 0.0009 | 0.002 | 0.0009 | 0.002 |

When comparing sampling methods, choice of parameters is very important, since each method has its own diversity-coherence trade-off characteristics. Without proper controls, it is impossible to tell whether the performance gap between two heuristics might be closed by simply adjusting the parameter of the worse-performing method. To remedy this, we control for parameter choice by matching parameters of compared methods based on how conservative they are with respect to human text. In particular, for each vanilla threshold sampling method $x$, we choose the BA-$x$ parameter that rejects the same proportion of tokens from a human corpus. Table 4 illustrates how we measure this *human-text rejection rate* (HRR). In our experiments, measure HRR by sampling 10,000 tokens with their prefixes from Open Web Text and calculating the proportion of the tokens that are accepted by a sampling method with a given parameter.

Table 4: With a hyperparameter of 0.002, $\epsilon$-sampling would have a human-text rejection rate of 1/5 on this text.

| Token | I'm | the | problem, | it's | me. |
|---|---|---|---|---|---|
| Probability | 0.02 | 0.3 | 0.01 | 0.001 | 0.3 |

Figure 8 gives the sampling parameters as a function of HRR. As HRR approaches zero, parameters become more permissive, i.e., nucleus approaches one, $\eta$ and $\epsilon$ approach zero, in order to accept more tokens. We observe that as HRR increases, BAT parameters are consistently more conservative than their vanilla counterparts since BAT methods sample tokens beyond the threshold. In the case of BA-$p$, the parameter maxes out around 28% HRR, meaning that it cannot reject more than 28% of human tokens.

### F.1  HUMAN EVALUATION

Annotators are paid $1 USD per annotation, and each annotation takes on average less than 2 minutes. Figure 9 provides the exact instructions and layout given to the annotators.

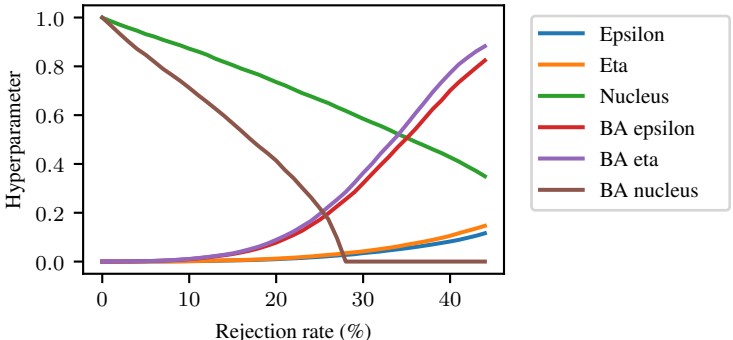

Figure 8: Truncation sampling parameters for various methods by HRR, the proportion of human-text the sampling methods reject.

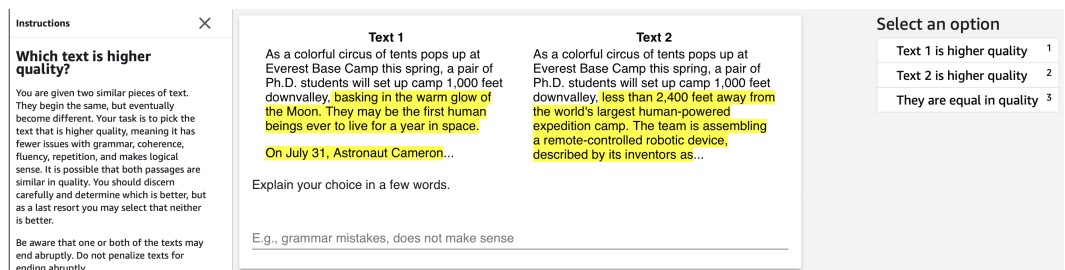

Figure 9: An example of the interface and instructions shown to human annotators.

## G    TRUNCATED LANGUAGE MODEL DISTRIBUTIONS ARE HIGH-RANK

We motivated truncation sampling as helping to correctly discard tokens that are not in the support of the true distribution $p^*$ when those errors are due to the low-rank nature of language models' distributions. In this additional experiment, we show that the post-truncation conditional distribution matrix $A$ is *high-rank* relative to the pre-truncation distribution.

We run the GPT2-XL model on samples of OpenWebText, concatenate the conditional log-distributions $\log \hat{p}$ for each prefix, and compute the rank of the resulting matrix. This becomes a rather large matrix, since each $\log \hat{p}$ is in $\mathbb{R}^{50257}$, so we are limited in the number of prefixes we can consider. Since the number of prefixes upper-bounds the estimated rank, and we cannot run, e.g., $50257$ prefixes, we plot the rank for various numbers of prefixes. We find that the GPT2-xl model, which has a hidden dimensionality of 1600, has rank that saturates at 1600, as expected. For truncation sampling strategies nucleus, $\eta$-sampling, and $\epsilon$-sampling, we find that the estimate of the rank continues to grow with the number of prefixes, far past 1600. See Table 10.

## H    MORE UNIT TESTS

We give the unit tests used in Figures 1 and 4 in tabular form (Tables 5-11).

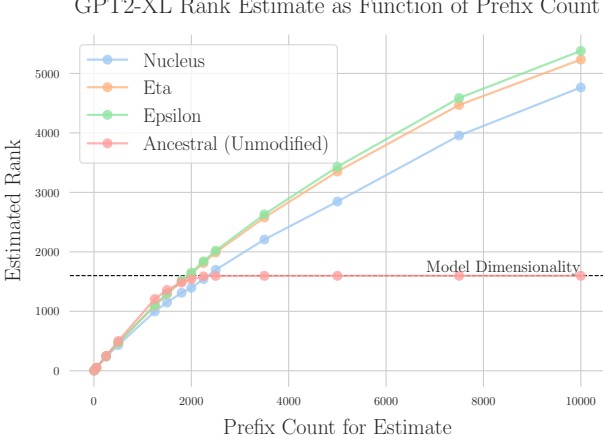

Figure 10: The estimated rank of the log-probability distributions of a model with truncation grow far past its hidden dimensionality; without truncation, the rank is constrained to the hidden dimensionality.

Table 5: A subset of the next-word distribution according to GPT2 for the context "`<|endoftext|>Taylor`". The last four columns denote whether each token is in the support of the titular strategies. Notice that BA-$\eta$ is able to accept good continuations like '` will`' and '` Hanson`' while excluding questionable continuations like '` Sw`' which likely has higher probability because of its embedding alignment with '` Swift`'.

| Rank | Prob | Token | BA Eta | Eta | Epsilon | Nucleus |
|---|---|---|---|---|---|---|
| 0 | 4.3e-01 | '` Swift`' | True | True | True | True |
| 3 | 2.1e-02 | '` is`' | True | True | True | True |
| 4 | 1.9e-02 | '` Hall`' | True | True | False | True |
| 29 | 2.5e-03 | '` Smith`' | True | True | False | True |
| 30 | 2.2e-03 | '` Sw`' | False | True | False | True |
| 31 | 2.2e-03 | '` K`' | True | True | False | True |
| 35 | 2.0e-03 | '` Miller`' | True | True | False | True |
| 36 | 2.0e-03 | '` Wilson`' | True | True | False | False |
| 38 | 1.9e-03 | '` will`' | True | True | False | False |
| 39 | 1.9e-03 | '` "`' | False | True | False | False |
| 40 | 1.9e-03 | '` says`' | True | True | False | False |
| 41 | 1.7e-03 | '` Hanson`' | True | False | False | False |
| 42 | 1.7e-03 | '` D`' | False | False | False | False |
| 43 | 1.7e-03 | '` Lew`' | True | False | False | False |
| 44 | 1.7e-03 | '` Hicks`' | True | False | False | False |
| 45 | 1.6e-03 | '` St`' | False | False | False | False |
| 46 | 1.6e-03 | '` C`' | False | False | False | False |
| 47 | 1.6e-03 | '` Wood`' | True | False | False | False |
| 50 | 1.5e-03 | '` Hein`' | True | False | False | False |
| 51 | 1.5e-03 | '` J`' | False | False | False | False |
| 60 | 1.2e-03 | '` Lee`' | False | False | False | False |
| 61 | 1.1e-03 | '` Kits`' | True | False | False | False |
| 62 | 1.1e-03 | '` Martin`' | False | False | False | False |
| 81 | 8.4e-04 | '` also`' | False | False | False | False |

Table 6: '`<|endoftext|>My name`'

| Rank | Prob | Token | BA Eta | Eta | Epsilon | Nucleus |
|---|---|---|---|---|---|---|
| 0 | 9.6e-01 | '` is`' | True | True | True | True |
| 1 | 3.2e-02 | '`'s`' | True | True | True | False |
| 2 | 1.2e-03 | '` was`' | True | False | False | False |
| 3 | 1.0e-03 | '`,`' | False | False | False | False |
| 4 | 7.7e-04 | '` isn`' | True | False | False | False |
| 5 | 4.7e-04 | '` Is`' | True | False | False | False |
| 6 | 4.1e-04 | '` and`' | False | False | False | False |
| 22 | 5.1e-05 | '` IS`' | False | False | False | False |

Table 7: '`<|endoftext|>My name is`'

| Rank | Prob | Token | BA Eta | Eta | Epsilon | Nucleus |
|---|---|---|---|---|---|---|
| 0 | 1.1e-02 | ' David' | True | True | False | True |
| 20 | 5.0e-03 | ' Adam' | True | True | False | True |
| 1156 | 1.3e-04 | ' Ily' | True | True | False | False |
| 1167 | 1.3e-04 | ' Curt' | True | True | False | False |
| 1168 | 1.3e-04 | ' Sk' | False | True | False | False |
| 1169 | 1.3e-04 | ' Stewart' | False | True | False | False |
| 1170 | 1.3e-04 | ' Avery' | True | True | False | False |
| 1175 | 1.3e-04 | ' Aud' | True | True | False | False |
| 1176 | 1.3e-04 | ' Eb' | False | True | False | False |
| 1177 | 1.3e-04 | ' Brock' | False | True | False | False |
| 1178 | 1.3e-04 | ' Franc' | True | True | False | False |
| 1184 | 1.3e-04 | ' Mercedes' | True | True | False | False |
| 1185 | 1.3e-04 | ' JJ' | True | False | False | False |
| 1194 | 1.2e-04 | ' Sebast' | True | False | False | False |
| 1195 | 1.2e-04 | ' Di' | False | False | False | False |
| 1196 | 1.2e-04 | ' Maxwell' | True | False | False | False |
| 1205 | 1.2e-04 | ' Mand' | True | False | False | False |

Table 8: '`<|endoftext|>The capital of of the USA is Washington D.C. The capital of India is New Delhi.  The capital of the UK is London.  The capital of Ghana is`'

| Rank | Prob | Token | BA Eta | Eta | Epsilon | Nucleus |
|---|---|---|---|---|---|---|
| 0 | 9.9e-01 | ' Acc' | True | True | True | True |
| 1 | 6.8e-03 | ' Ab' | True | False | False | False |
| 2 | 6.7e-04 | ' Kum' | True | False | False | False |
| 3 | 2.6e-04 | ' Con' | False | False | False | False |
| 4 | 2.3e-04 | ' Ghana' | True | False | False | False |
| 5 | 1.4e-04 | ' Abu' | False | False | False | False |
| 22 | 1.2e-05 | ' Tem' | False | False | False | False |

Table 9: '`<|endoftext|>Donald`'

| Rank | Prob | Token | BA Eta | Eta | Epsilon | Nucleus |
|---|---|---|---|---|---|---|
| 0 | 9.4e-01 | ' Trump' | True | True | True | True |
| 1 | 1.4e-02 | ' J' | True | False | False | False |
| 2 | 6.2e-03 | ' Glover' | True | False | False | False |
| 3 | 1.9e-03 | ' Sterling' | False | False | False | False |
| 22 | 3.1e-04 | ' Donald' | False | False | False | False |

Table 10: '`<|endoftext|>The`'

| Rank | Prob | Token | BA Eta | Eta | Epsilon | Nucleus |
|---|---|---|---|---|---|---|
| 0 | 1.3e-02 | ' first' | True | True | False | True |
| 20 | 2.9e-03 | ' American' | True | True | False | True |
| 4338 | 3.5e-05 | ' RNC' | True | True | False | False |
| 4340 | 3.5e-05 | ' poet' | True | True | False | False |
| 4354 | 3.5e-05 | ' BE' | True | True | False | False |
| 4357 | 3.5e-05 | ' inevitable' | True | True | False | False |
| 4358 | 3.5e-05 | ' hackers' | True | False | False | False |
| 4359 | 3.5e-05 | ' Bright' | True | False | False | False |
| 5232 | 2.8e-05 | ' PBS' | False | False | False | False |
| 5233 | 2.8e-05 | ' Grammy' | False | False | False | False |

Table 11: '<|endoftext|>The feeling!  The feeling!  The feeling!  The
feeling!  The feeling!  The feeling!  The feeling!  The feeling!
The feeling!  The feeling!  The feeling!'

| Rank | Prob | Token | BA Eta | Eta | Epsilon | Nucleus |
|---:|---:|---|---|---|---|---|
| 0 | 9.5e-01 | ' The' | True | True | True | True |
| 1 | 2.3e-02 | '\n' | True | False | True | False |
| 2 | 2.6e-03 | ' THE' | True | False | False | False |
| 3 | 2.2e-03 | '\n\n' | False | False | False | False |
| 4 | 1.8e-03 | ' I' | False | False | False | False |
| 5 | 9.9e-04 | 'The' | True | False | False | False |
| 6 | 6.3e-04 | ' It' | False | False | False | False |
| 22 | 1.3e-04 | ' My' | False | False | False | False |

