# OpenReview forum: "Closing the Curious Case of Neural Text Degeneration"
_ICLR.cc/2024/Conference — ICLR 2024 poster_

### Official Review · Reviewer_4bep · 2023-10-18

**Soundness:** 3 good
**Presentation:** 3 good
**Contribution:** 3 good
**Rating:** 8
**Confidence:** 4

**Summary:**

This paper aims to develop a more precise sampling strategy for language models, specifically focusing on addressing errors arising from the softmax bottleneck. The authors establish two sufficient conditions, under certain assumptions, to ensure that sampled tokens belong to the true distribution's support. The first condition (Corollary 1) leads to the threshold sampling algorithm, providing a direct explanation for its success. Assuming the loss to be cross-entropy, the second condition imposes a linear constraint that can be solved with a linear programming optimizer.Combining both conditions, the proposed basis-aware threshold (BAT) sampling outperforms its threshold-based counterparts for low-entropy open-ended text generation.

**Strengths:**

1. The proposed approach is innovative and theoretically-grounded.
2. This paper brings new insights to the community. The theoretical concepts discussed in this paper are previously ignored but seems to be important.

**Weaknesses:**

1. Basis-aware sampling is specifically designed for models trained using cross-entropy loss. However, not all language models meet this criterion. For instance, LLMs fine-tuned with RLHF do not adhere to this condition.
2. Theorem 2 and Corollary 2 provide sufficient but unnecessary conditions for proving tokens are in the true support. Therefore, the induced sampling algorithm may also discard tokens in the support of true distribution, leading to biased sampling.
3. In Corollary 1, the statement "threshold-based truncation sampling correctly discards all tokens that are not in the support of p*" is not precise, as it may also incorrectly discard tokens that are in the support of p*. A more precise phrasing would be: "all tokens that are not in the support of p* will be discarded by threshold-based truncation sampling."
4. The relationship between the softmax bottleneck and text degeneration phenomena has not been verified. It remains unclear whether text degeneration is directly caused by the softmax bottleneck, or if increasing the dimensionality (d) beyond the vocabulary size (v) would effectively resolve the issue.

**Questions:**

None

---

> ### Author Response · Authors · 2023-11-16
>
> Thank you for your encouraging words! We are also excited by our approach and agree that this topic has been overlooked and underexplored, despite its centrality to generative language models. We hope we can address some of your concerns below.
>
> > Basis-aware sampling is specifically designed for models trained using cross-entropy loss. However, not all language models meet this criterion. For instance, LLMs fine-tuned with RLHF do not adhere to this condition.
>
> We chose to focus on cross-entropy loss since it is the most popular method used to pretrain base language models. We agree that it would be worthwhile to consider the implications of alternative loss functions, though these are not guaranteed to give the “nice” linear constraints that we derive from cross entropy.
>
> > Theorem 2 and Corollary 2 provide sufficient but unnecessary conditions for proving tokens are in the true support. Therefore, the induced sampling algorithm may also discard tokens in the support of true distribution, leading to biased sampling.
>
> This is true. By design, these methods prioritize precision over recall, though the degree of conservativeness can be refined by tuning the hyperparameter $\tau$. This is actually a core contribution of our work: threshold-based methods can guarantee only in-support tokens are sampled, but at the cost of being overly conservative and rejecting good tokens. Our method allows us to lessen this issue by taking full advantage of our knowledge of a source of model error. We cannot recover the true distribution, but BAT allows us to get closer than vanilla thresholding.
>
> > In Corollary 1, the statement "threshold-based truncation sampling correctly discards all tokens that are not in the support of p*" is not precise, as it may also incorrectly discard tokens that are in the support of p*. A more precise phrasing would be: "all tokens that are not in the support of p* will be discarded by threshold-based truncation sampling."
>
> This is indeed what we meant. We believe the confusion may have come from the word “correctly” which we will remove.
>
> > The relationship between the softmax bottleneck and text degeneration phenomena has not been verified. It remains unclear whether text degeneration is directly caused by the softmax bottleneck, or if increasing the dimensionality (d) beyond the vocabulary size (v) would effectively resolve the issue.
>
> Prior work [1,2] offers evidence that the softmax bottleneck (SMB) indeed negatively impacts language model performance, both in theory and in practice, and it has been shown that “breaking” the SMB can lead to better performance. A question that remains is whether (and when) this effect will have an impact on generation, since heuristics like threshold sampling may already mitigate the SMB effects to some degree. Our pilot experiments with low-entropy decoding seem to suggest that the SMB may still have a negative impact on low-entropy decoding that can be mitigated by BAT sampling.
>
> [1] Yang et al., Breaking the Softmax Bottleneck: A High-rank RNN Language Model.
>
> [2] Yang et al., Mixtape: Breaking the Softmax Bottleneck Efficiently.

---

### Official Review · Reviewer_8Qv3 · 2023-11-01

**Soundness:** 3 good
**Presentation:** 3 good
**Contribution:** 3 good
**Rating:** 8
**Confidence:** 2

**Summary:**

This paper provides a theoretical analysis of why decoding from language models with truncation sampling works well and provides a new decoding strategy called BAT-sampling. The gist is that if you truncate a sufficiently large portion of the distribution, then truncation sampling will avoid any tokens which are not in the support of the true language distribution. However, this may result in throwing out tokens which *are* in the support of the distribution. The paper then proposes BAT-sampling, which is an LP-based method that can determine which tokens are and are not in the support of the true distribution, even if they are "out of order" in terms of the probability assigned (i.e., the method can determine that lower-probability tokens are in the support of the distribution even when higher probability tokens are not). The paper concludes with a set of experiments, including an impressive discussion of speedups for BAT-sampling, as well as some (very slight) improvements over existing methods in certain conditions.

**Strengths:**

This is a great analysis paper, providing an interesting explanation for why truncation sampling works so well in language model decoding. The paper's motivation is clear and well-written. The fact that BAT can determine that some tokens have nonzero true support, even though they are assigned less probability than others which are not in the support of the true distribution, is a surprising and compelling result. Leveraging the softmax bottleneck is a clever trick here and one that will be unexpected to most readers in NLP.

I expected BAT to be computationally infeasible to run in practice due to its dependence on an LP-solver at each tilmestep of decoding. However, the speedups in the "Basis-aware threshold sampling in practice" (namely, using a decomposition of the softmax matrix and only relying on BAT when a token under the threshold probability is chosen) seem reasonable and compelling, and the amortized cost of 0.1s/token, while slow, is not infeasible for certain classes of applications.

The experiments, although not particularly compelling as a reason to start using BAT sampling in practice, seem reasonable and sufficiently thorough. In particular, the analysis of performance as more constraints are added back (after the SVD) is very clear. In contrast, I did not find the "BAT outperforms all other methods for GPT-2-Large" paragraph very compelling given that BAT is not the best-performing model on any other model size.

**Weaknesses:**

The primary weakness seems to be the performance of BAT compared to other methods. Despite its theoretical justification, it does not clearly outperform other sampling approaches (Figure 5). Although there is a preference for BAT to eta-sampling shown in Figure 6 and Table 1, this preference is very slight and the comparison is only between two sampling methods. However, I do not see this weakness as a legitimate reason to reject the paper, since its main contribution seems to be analysis and theoretical understanding of existing decoding algorithms.

**Questions:**

1. Based on the figures (1,4), it seems like BAT is rejecting a lot of tokens corresponding to partial words. Out of curiosity: is this true, and do you have any insights into why this happens, or other qualitative insights into what tokens tend to get accepted/rejected?

---

> ### Author Response · Authors · 2023-11-16
>
> Thank you very much for your perspective! We worked hard to make our paper clear, both in writing and mathematically. We were equally surprised and excited by the idea of recovering information about the true support using the softmax bottleneck. We put a lot of effort into making BAT computationally feasible and are pleased with the speedups we achieved. We agree that our pilot studies, while not fully convincing as a reason to immediately switch to BAT, offer indications that BAT could prove helpful in certain generations settings and offer some empirical and anecdotal evidence that our proposed method warrants further development and investigation.
>
> > The primary weakness seems to be the performance of BAT compared to other methods. Despite its theoretical justification, it does not clearly outperform other sampling approaches (Figure 5). Although there is a preference for BAT to eta-sampling shown in Figure 6 and Table 1, this preference is very slight and the comparison is only between two sampling methods. However, I do not see this weakness as a legitimate reason to reject the paper, since its main contribution seems to be analysis and theoretical understanding of existing decoding algorithms.
>
> We agree that our pilot experiments show only preliminary evidence for the effectiveness of our methods, and we are glad that you share our view that these results are supplementary, rather than the primary contribution of our paper, which is the theoretical explanation and mathematically grounded algorithmic proposal.
>
> Questions:
>
> > Based on the figures (1,4), it seems like BAT is rejecting a lot of tokens corresponding to partial words. Out of curiosity: is this true, and do you have any insights into why this happens, or other qualitative insights into what tokens tend to get accepted/rejected?
>
> This is a great observation! Our leading hypothesis for why these partial words are rejected (e.g., `Swift` accepted, but `S` and `Sw` rejected) is that they have high embedding similarity with the full word. Tokens whose embeddings (in $W$) have high dot-product similarity with the model’s hidden state receive the most probability mass. In our example, the model output a hidden state with high dot-product similarity with `Swift`, and consequently tokens like `Sw` that have similar embeddings to Swift also have high dot-product similarity with the hidden state, resulting in them getting probability mass.
>
> Using our method, in order to identify a target token for truncation, it must be possible to transfer all the probability mass on the target token to other tokens while maintaining the constraints of the linear program. The $W^\top\hat{p}=W^\top p$ constraint specifies that the probability mass must be transferred to tokens with similar embeddings to the target token, and the $p_j\leq\hat{p}_j\exp(\delta)$ constraint limits the amount of mass that can be transferred to any one token, where the higher probability a token is, the more probability mass can be added to it. Therefore, a token is most likely to be truncated if there is another token with high probability and high embedding similarity to it.
> In our example, the probability mass on `Sw` embedding can be transferred to `Swift` which has high probability and (likely) high embedding similarity.

---

### Official Review · Reviewer_Hq1N · 2023-11-02

**Soundness:** 3 good
**Presentation:** 3 good
**Contribution:** 3 good
**Rating:** 6
**Confidence:** 4

**Summary:**

The paper proposes a theoretical understanding of truncation sampling methods. The authors proceed to devise a novel sampling strategy, building upon the approximation error incurred by the softmax output bottleneck. Essentially, the idea is to assume that a token has a non-zero probability under the true distribution, not only if it has a non-zero probability under the predicted distribution (which is a source of overestimation errors) but also if it is non-zero under all distributions that "map back" to the hidden state by taking the transpose of the output embedding matrix. Intuitively, this measures whether a token has a non-zero probability by chance, i.e. if its information can be conveyed by any combination of other tokens while mapping back to the hidden state. This is formalized in the paper by assuming that the hidden state is the minimizer of the cross-entropy loss with respect to the true distribution. This insight serves to devise a novel sampling strategy that can sample low-probability next tokens, which differs from current approaches that rely on thresholding.

**Strengths:**

- Nice idea and analysis
- Well written / clear
- Shines new insights on a well-studied problem and could lead to more promising sampling methods

**Weaknesses:**

- Results are rather weak, efficacy of the method still remains to be demonstrated (minor)
- An pseudo-code / algorithm box with the practical implementation of BA is needed in the main paper (minor)
- Unclear whether the method will help for larger models or for models where the approximation errors (under-estimation / over-estimation) are small (kinda major).

**Questions:**

Thank you for the efforts in writing a clear and enjoyable paper.

Main questions:
- Can you include a pseudo-code of the final BA implementation in the main paper?

- Regarding Eq. 3: what I am going to propose is a bit dirty but would it be possible at each step to minimize (W^T p - W^T \hat p)^2, wrt to p with a sparsity constraint (e.g. l1) and the range constraints, and reject all tokens for which |p| = 0? It might not derive from the theory but it might capture the overall idea? just wondering.

- Concerning the results: there isn't much of a pattern in the MAUVE results if I look at the improvements of BA across model scales. Isn't  BA expected to help more with smaller scales given that the approximation error might be bigger?

- Main problem: what happens with bigger models? given that the approximation error will be smaller, would your method still help?

Nitpicks:
- It might be clearer to re-introduce the \epsilon and \eta baselines in the experiments. I struggled a bit to remember given that they are just introduced in the background section.
- Eq. 10 in the appendix is missing a parenthesis.

I would love to give a 7, but I can't (I have to choose between 6 and 8 now). I will give a 6 for now, and wait for authors responses with the will to increase my scores if more details / addition to the papers are given :-)

---

> ### Author Response · Authors · 2023-11-16
>
> We are very grateful for your kind words, as we worked hard to make our writing and math clear, and are very excited about this new perspective and the resulting insights. We hope that we can sufficiently address your concerns.
>
> > Results, efficacy of the method still remains to be demonstrated (minor)
>
> We agree that our pilot experiment results are largely preliminary and will benefit from further investigation in future work. We take the view that our experimental findings are secondary and meant to supplement our primary theoretical results and method proposal.
>
> > Unclear whether the method will help for larger models or for models where the approximation errors (under-estimation / over-estimation) are small.
>
> One important implication of our results is that larger models do not automatically avoid under/over-estimation errors, since the errors caused by the softmax bottleneck (SMB) depend not on the model size, but on the discrepancy between the hidden size and the vocabulary size. A larger model with a proportionally larger vocabulary size will suffer the same bottleneck effects. By the same argument, a model with a small vocabulary (i.e., close to equal hidden and vocab sizes) could avoid the SMB altogether, no matter the size. Unfortunately, we do not see many models of this type in common use.
>
> One size-related investigation we would like to pursue in future work is the hypothesis that our method will become **more** effective for larger model sizes. We hypothesize that some of our results were weaker than expected since our method primarily addresses SMB-related errors. If the model predicts the next token incorrectly, none of the considered truncation methods (including ours) will be able to correct the error. This issue should be reduced with larger models, since larger models should have fewer modeling errors, leaving only SMB errors. We see some anecdotal evidence for this where, for instance, BA-$\eta$ becomes stronger compared to $\eta$ sampling as model size increases (Figure 5).
>
> > Regarding Eq. 3: what I am going to propose is a bit dirty but would it be possible at each step to minimize (W^T p - W^T \hat p)^2, wrt to p with a sparsity constraint (e.g. l1) and the range constraints, and reject all tokens for which |p| = 0? It might not derive from the theory but it might capture the overall idea? just wondering.
>
> This idea is worth exploring! At first glance, the major difference with this approach is that it would require all truncated tokens to have 0 probability simultaneously, whereas our method only requires a solution with each truncated token having 0 probability independently. The advantage of your proposed method would be a faster runtime, with the drawback that the list of truncated tokens might be incomplete. Perhaps there is a middle ground, where we iteratively find solutions, each time finding multiple tokens to truncate, then removing the sparsity constraint from these tokens for the next iteration so we can find new potentially-0-probability tokens.
>
> > Concerning the results: there isn't much of a pattern in the MAUVE results if I look at the improvements of BA across model scales. Isn't BA expected to help more with smaller scales given that the approximation error might be bigger?
>
> See our response to your concern above: there may be multiple effects here, such as the rate at which the model makes prediction mistakes independent of the SMB, which would confound the trend because these types of mistakes would become less frequent at larger model sizes.
>
> > What happens with bigger models? given that the approximation error will be smaller, would your method still help?
>
> Again, we refer to our size-related responses above, with an additional perspective: we expect that small models will continue to play a role in language generation, given the ease of deployment in edge devices, as well as cheaper inference costs. Even if our method does not provide benefits for larger models, smaller models will continue to suffer from a SMB and our method (or some modification of it) will remain relevant.
>
> > Nitpicks: It might be clearer to re-introduce the \epsilon and \eta baselines in the experiments. I struggled a bit to remember given that they are just introduced in the background section. Eq. 10 in the appendix is missing a parenthesis. Can you include a pseudo-code of the final BA implementation in the main paper?
>
> Thank you for pointing these out, we update accordingly in our revision.

---

### Official Review · Reviewer_pgkn · 2023-11-03

**Soundness:** 3 good
**Presentation:** 4 excellent
**Contribution:** 3 good
**Rating:** 8
**Confidence:** 4

**Summary:**

This paper aims to give a formal justification for why truncation based sampling approaches work well in language generation. They link this phenomenon to the softmax bottleneck—the problem that the final linear layer in a neural network often bottlenecks the expressivity of the model. Explicitly, given the difference between the hidden (embedding) dimension and the vocabulary size, the final linear transformation before the softmax projection can only perform a low-rank projection. The authors claim that the resulting approximation to the target distribution is likely the source of model errors that leads to the “unreliable tail” probabilities observed by prior work. The authors develop an algorithm for uncovering which tokens are necessarily in the support of the target distribution, and propose to use this algorithm as the basis for a truncation sampling method. They provide empirical results (including human evaluations) when using this method.

**Strengths:**

* The work offers a theoretical explanation for why certain ad-hoc methods used during language generator decoding work well. This is a valuable insight to the NLG community
* The work then develops a sampling algorithm based on this theoretical explanation

**Weaknesses:**

* The theoretical portion of the paper is at times difficult to understand due to notational choices and lack of specificity (for example, switching between individual token probabilities ). This is particularly important since the theoretical portion is the main contribution of the work
* The method does not appear to have empirical performance benefits and is computationally expensive, making it impractical
* There lacks robust empirical justification of the hypothesis. Figure 7, which is intended to show that including more of the original optimization problem constraints lead to better results, only consists of 3 points, which hardly feels like enough evidence to claim a “trend”.
* The terminology of the “true” distribution is perhaps misleading. I personally think that something like the “aggregate” distribution or the “data-generating” distribution would be more accurate
* A small point: the intro of section 4 has some grammatical errors

**Questions:**

* Since the matrix W is static, can (3) not be solved for all elements of the vocabulary that meet the desired constraint ahead of time?
* Why is typical sampling omitted from the discussion of truncation-based sampling methods? It is arguably the most similar to the proposed method since it likewise “is able to discard higher-probability tokens while keeping… lower-probability tokens.” On a similar note, I don’t understand footnote 3; I don’t think it actually describes what is done by locally typical sampling
* The phrasing of footnote 1 is strange. Specifically, the statement “setting p∗ to be the 1-hot vector indicating the gold token” is underspecified. I imagine that this is referring to the conditional distribution for a particular prefix
* In theorem 1, these factors are the collective probability over/underestimation across all tokens, right? This then implies that no individual token probability can exceed these bounds. This logic should be made more explicit (the current notation is vague)
* I don’t think that the softmax function satisfies the additivity property required of a “linear map.” Could you please elaborate on this claim at the top of page 6?
* On page 5, what is the concrete distinction between low vs. high quality tokens? Is this another way of saying in vs. out of the true distribution? It would be helpful to change the language here to align with the other terminology used by the paper
* The informal description about the practical implementation of basis-aware sampling is confusing. For example, what does “discarding the majority of the constraints” refer to?
* Given the observation that BAT sampling performs more strongly in lower entropy settings, a logical next step would be to see how it performs in translation or summarization, where historically, sampling has not led to the best results.
* It is perhaps more accurate to call the projection by W a linear layer instead of softmax layer, since the low-rank approximation is tied to this linear transformation, not the use of the softmax as a projection function. Further, are there insights into how the nature of these results will change with alternative projection functions, like the sparsemax?
* Can these principles be used to explain the degeneracy that happens when selecting high probability tokens, i.e., during greedy decoding?
* How do these results align with other work that has tried to explain why truncation methods work well in practice, such as [1]?

[1] Meister et. al. 2023. On the Efficacy of Sampling Adapters.

---

> ### Author Response · Authors · 2023-11-16
>
> Thank you for your kind words and appreciating our contributions. We appreciate your feedback, insights, and overall engagement with our paper.
>
> > Notational choices
>
> We strived to follow the ICLR style guidelines for notation, and also to be as consistent and specific as possible. We caught a few notational errors in our revision too. We would appreciate more feedback on which parts are confusing.
>
> > Performance benefits for computational expense
>
> We address the overhead at the end of Sec. 4: the overall addition is $\approx0.1$ seconds per token, which is acceptable for many applications. Additionally, this method can be further optimized in the future.
>
> > Figure 7 only 3 points
>
> Each point in Fig 7 is a mean over multiple seeds (fill region indicates standard deviation), which helps establish the trend. While our sampling method could benefit from refinement (see Section 5.2), we contend that our MAUVE experiments when taken together with our qualitative analyses (Figure 4) *do* indicate that the softmax bottleneck is a significant source of model errors, especially in low-entropy generation settings. While BAT does not particularly stand out in Fig 5, no other truncation method is consistently superior either. We believe that this is because high-entropy generations are generally similar across truncation methods.
>
> We would like to highlight that these empirical analyses are only pilot studies, and that the main contributions of this paper are a theoretical explanation for threshold-based truncation sampling efficacy, and a mathematically-grounded method for mitigating distribution errors from the softmax bottleneck.
>
> > The “true” distribution
>
> We address this in Footnote 1, where we refer to the distribution “from which internet text is implicitly sampled”, which we believe is equivalent to the “data-generating” distribution.
>
> > Solve (3) ahead of time?
>
> This would be desirable, but we cannot pre-solve since $\hat{\boldsymbol{p}}$ is different for each context. Pre-compilation of the program could speed up solving.
>
> > Typical sampling, footnote 3
>
> See the new Footnote 3. We focus less on typical sampling since it is markedly different from the other truncation schemes in that it does not attempt to recover the support of true distribution.
>
> > The phrasing of footnote 1
>
> We mean that the model is trained to minimize cross-entropy with a surrogate distribution that puts all probability mass on the gold token. Since this is confusing and not particularly relevant, we remove this.
>
> > Softmax function a “linear map.”
>
> This is indeed unintuitive. See our newly added Appendix C for a short explanation. For more, see https://golem.ph.utexas.edu/category/2016/06/how_the_simplex_is_a_vector_sp.html.
>
> > Informal description of practical implementation confusing.
>
> “Discarding the majority of the constraints” means that we use an approximation instead of the full $\boldsymbol{W}$ matrix to make the linear program faster. See revision.
>
> > BAT for translation or summarization
>
> This is a great future work direction, which we are pursuing!
>
> > Call the projection by W a linear layer
>
> You are right about this: the bottleneck has nothing to do with the softmax function and everything to do with the linear projection parameterized by the “embedding” matrix $\boldsymbol{W}$. Our terminology is for historical reasons (see new Footnote 5).
>
> > Insights into sparsemax?
>
> Great question! We focus exclusively on the softmax function since it is the most commonly used, but the principles we use **may** be applicable to the sparsemax function, since it preserves some (though not all) the linear dependencies between token embeddings.
>
> > Can these principles explain degeneracy from selecting high probability tokens?
>
> We suspect that degeneracy in greedy decoding stems from sampling from an unnatural distribution. One possible explanation for why our method does better in low-entropy (close-to-greedy) decoding may be that there are some non-greedy tokens that our method refuses to truncate ever. As an example, in the final frame of Figure 4 most threshold methods continue the repetition, but BAT allows other tokens to be sampled.
>
> > How do these results align with other work, such as (Meister et. al., 2023)?
>
> Meister et al. (2023) approach the question of “why do truncation methods work” by hypothesizing that threshold sampling reprioritizes precision over recall. In some respects, our explanation presupposes the Meister et al. hypothesis then proves guarantees on the mechanism by which threshold sampling achieves high precision. Other work [2] has also made the assumption that threshold sampling aims to avoid zero-probability tokens. We add Meister et al. to our revision.
>
> > Other minor comments
>
> See revision.
>
> [1] Yang et al., Breaking the Softmax Bottleneck: A High-rank RNN Language Model.
> [2] Hewitt et al. Truncation sampling as language model desmoothing.

---

> > ### Comment · Reviewer_pgkn · 2023-11-22
> > **Response**
> >
> > Thank you for the clarifications! Several of my concerns are addressed. I have raised my score to an 8 accordingly.

---

### Author Response · Authors · 2023-11-16

Based on your feedback, we have made edits to our submission and uploaded a rebuttal revision. The changed text in the main document has been colored red for your convenience.

We would like to thank all of our reviewers for their enthusiasm for our work and the great questions! We enjoyed reading all of your ideas and inputs, and appreciate the depth of your engagement with our paper. We hope we can properly address the concerns you raise.

One recurring comment we received was concern over the strength of the empirical results in our pilot studies. As reviewers recognized (and as we note in the paper) our theoretical findings constitute the main contribution, and our pilot experiments serve more as supplementary, to explore and validate our proposed method. We indeed do not find sufficient evidence to recommend replacing existing truncation sampling algorithms with BAT. On the other hand, in our attempt to present an honest evaluation of BAT, we find that it indeed seems to provide benefits over other sampling methods in low-entropy generation settings. We interpret this as an indication that our method warrants further investigation and refinement and may prove useful for applications involving close-ended generation tasks, such as translation and summarization (as Reviewer pgkn points out).

Once again, thank you for your thorough reviews. We hope our individual replies will help address your remaining concerns, and we are excited to discuss more as well if you have any followup questions.

---

### Meta-Review · Area_Chair_UfUK · 2023-12-12

**Metareview:**

This is a nice paper that examines why truncation-based sampling methods work well when decoding from LMs. They theorize that the softmax bottleneck results in an often flawed approximation of the the target token distribution. Then, they develop a method that does not rely on thresholding but rather identifies tokens that are in the support of this target distribution while truncating the rest. Interestingly, they identify several tokens in the support that are lower probability than other tokens not in the support. Reviewers (and this AC!) agree that the paper's findings are interesting and impactful. The main weakness is certainly the ineffectiveness of the proposed decoding algorithm compared to standard alternatives (and its general slowness); in general, however, I think this is outweighed by the quality of the analysis.

**Justification For Why Not Higher Score:**

I do think the lack of compelling results with the proposed decoding algorithm makes the paper a *little* bit underwhelming.

**Justification For Why Not Lower Score:**

That said, the main contribution of the paper could be significantly impactful for future work on decoding and LM architecture development, so it definitely should not be rejected.

---

### Decision · Program_Chairs · 2024-01-16

Accept (poster)